# Sensing Technologies for Measuring Grain Loss during Harvest in Paddy Field: A Review

**Muhammad Isa Bomoi** [1,2,*] **, Nazmi Mat Nawi** [1,3,4] **, Samsuzana Abd Aziz** [1,3]
**and Muhamad Saufi Mohd Kassim** [1,3,4]

1    Department of Biological and Agricultural Engineering, Faculty of Engineering, University Putra
     Malaysia (UPM), Serdang 43400, Malaysia; nazmimat@upm.edu.my (N.M.N.);
     samsuzana@upm.edu.my (S.A.A.); saufi@upm.edu.my (M.S.M.K.)
2    Scientific Equipment Development Institute, Minna 920001, Nigeria
3    SMART Farming Technology Research Centre, Faculty of Engineering, University Putra Malaysia (UPM),
     Serdang 43400, Malaysia
4    Institute of Plantation Studies, University Putra Malaysia (UPM), Serdang 43400, Malaysia
*    Correspondence: mibomoi03@gmail.com

**Abstract:** A combine harvester has been widely employed for harvesting paddy in Malaysia. However, it is one of the most challenging machines to operate when harvesting grain crops. Improper handling of a combine harvester can lead to a significant amount of grain loss. Any losses during the harvesting process would result in less income for the farmers. Grain loss sensing technology is automated, remote, and prospective. It can help reduce grain losses by increasing harvesting precision, reliability, and productivity. Monitoring and generating real-time sensor data can provide effective combine harvester performance and information that will aid in analyzing and optimizing the harvesting process. Thus, this paper presents an overview of the conventional methods of grain loss measurements, the factors that contribute to grain losses, and further reviews the development and operation of sensor components for monitoring grain loss during harvest. The potential and limitations of the present grain loss monitoring systems used in combine harvesting operations are also critically analyzed. Several strategies for the adoption of the technology in Malaysia are also highlighted. The use of this technology in future harvesting methods is promising as it could lead to an increase in production, yield, and self-sufficiency to meet the increasing demand for food globally.

**Keywords:** combine harvester; grain; paddy; losses; sensor; monitoring; measurements

## 1. Introduction

Combine harvesters are widely utilized to harvest different kinds of grains [1]. The machine is one of the most difficult pieces of agricultural equipment to operate while harvesting grain crops [2]. Modern combines come in a variety of shapes and sizes. In a range of crop and field conditions, a combine harvester harvests and threshes all forms of grains such as rice, wheat, corn, soybeans, etc. [3]. It is important to keep combine harvester operating conditions and output high because of the urgency of farming and rice production [4]. Considering labor shortages, using a combine harvester increases labor productivity and reduces production costs. However, inappropriate crop conditions and machine settings cause adverse effects on the quantity and quality of rice grains.

In order to increase rice quality and quantity, the output of the combine harvester must be as efficient as possible in order to reduce grain losses. Combine harvesters perform the following operations when harvesting crops: stem cutting, seed detachment, threshing, separation procedures, as well as grain cleaning and collection in a tank. Among the processes involved, grain losses (such as header losses, threshing losses, separation losses, and cleaning losses) have become a major challenge [5].

One of the main obsessions in the direction of machine performance and loss control is grain losses caused by harvesting with a combine harvester [6]. After fulfilling the mechanization of harvesting, monitoring and controlling grain loss in the process of rice harvesting has become one of the most significant problems involved in intelligent harvesting [7]. Among the combine performance parameters, the grain loss rate is significant for combine harvesters [8]. During harvesting, grain losses represent a direct loss of income for the farmers [9]. Inappropriate technology and techniques result in 25–30% losses in rice yield, and to reduce these losses; the harvesting parameters need to improve for combine harvesters [10]. It is important to have reliable measures of these losses [11].

According to [12], a reasonable amount of grain losses should not reach a maximum of 3% of the total crop yield. Reference [13] reported a grain loss of 5.3 percent in the header combine harvester, exceeding the minimum limitations. As a result, more studies will be required to find characteristics and operating conditions that minimize grain loss and combine harvester loss. Reference [14] estimated 5% rice harvesting losses from the overall weight of harvested rice during combine harvesting.

One of the most important aspects of grain loss reduction is the impact of the combine harvester [15], and mechanical losses are inevitable during the working process of a combine harvester and can only be reduced in an appropriate way [16]. Inefficient harvesting by a combine harvester results in significant grain loss in the field. Importantly, inadequate control of operational parameters such as combine handling, environment, moisture, and crop physiological variables causes grain loss. The crop's physiological maturity is one of the factors to blame for the grain losses apart from machine factors [17]. It is estimated that about 80% of grain harvest losses occur at the combine's header, which can significantly impact the crop's net return on the harvest [18]. Reference [19] reported a 1.39% grain loss in their study, which led to income losses of up to USD 29.26 (RM 120.55) per hectare. This demonstrates that farmers incur significant losses during harvesting operations, resulting in a lower (decreased) profit.

It is very important to determine total grain losses qualitatively and quantitatively as well as the quality of harvested mass in a rice combine harvester, not only for economic calculation, determination of total yield, and the effects of the harvester but also for global food security concerns because any amount of grain losses could affect the world's feeding population [9]. Thus, the adoption of technology will benefit operators by facilitating proper machine handling to minimize grain losses during harvesting.

Typical grain loss sensors are made up of piezoelectric films and piezoelectric ceramics, which convert the impact of materials into an electrical signal. Different materials produce different response signals, and those raw electrical signals are processed through a series of steps, including amplification, filtration, wave rectification, envelope dithering, and amplification [20]. When utilized appropriately, grain loss sensors may be one of the most effective instruments for assisting an operator in determining how effectively his combine is functioning [21]. Improvements in the display of grain loss sensor signal data in meaningful units of absolute grain loss will assist operators and farm managers to make better economic choices and manage grain loss throughout the harvest season, respectively [22].

Specifically, the goal of this article is to examine the need and potential for grain loss control and describe the capabilities of emerging sensor technologies for grain loss measurement. The purpose of this paper is to identify and discuss the parameters that contribute to grain losses in combine harvesters and conventional methods of measuring grain losses, as well as the various types of sensors used to measure and monitor grain losses, and issues concerning the design, operation, and limitations of sensing technology used in combine harvesting operations.

## 2. Major Losses in Combine Harvester Operation

A combine harvester is a complex, self-propelled farm machine that performs multiple tasks. Before making any changes or adjustments to reduce losses, it is helpful to understand the processes that occur in the combine. The operation of a combine harvester as highlighted

in Figure 1 begins at the header unit, where cutting and gathering of the grains takes place. The material is subsequently carried into a threshing unit, where it is processed, cleaned, and separated (straw walker). The clean grain is transported to the storage tank while the straw and chaff are discharged onto the field. When the grain tank is filled, the grain is further discharged by an auger into a trailer or truck.

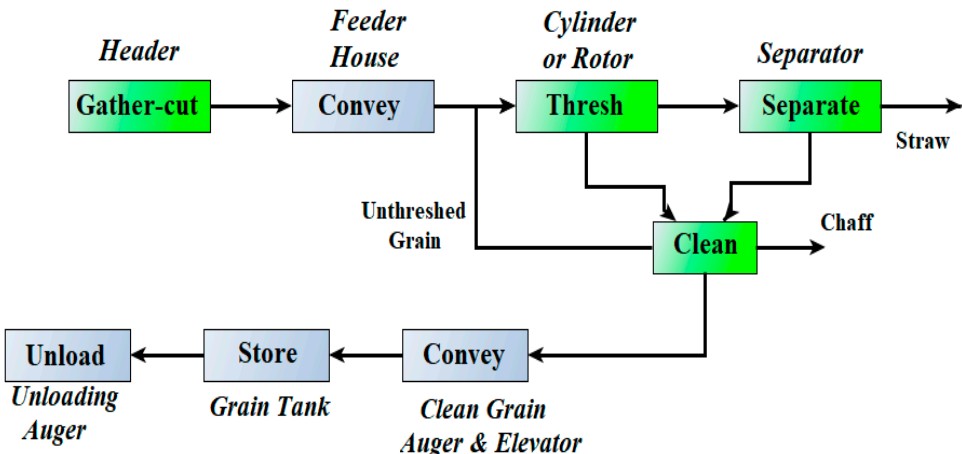

**Figure 1.** Combine harvester operation procedure highlighting where losses typically occur.

The loss of grain in the field caused by harvesters is classified into header losses (losses at the front of a harvester) and combining losses (losses at threshing, separation, and cleaning units) or gathering losses and processing losses. The following are different types of grain losses in the field before and during the combining of crops during harvesting operations:

### 2.1. Header Losses

Header losses occur during the gathering, which is the process of feeding the crop into the machine header prior to threshing [18]. Header losses can be expressed as kg/ha or as a percentage of the crop yield [23]. The losses caused by the cutter bar are considered important, as the header unit is the component that vigorously hits the panicles. The factors that affect header loss during the combine harvesting operation include reel height, cutter/header height, distance from the blade bar to the reel center, and the reel speed relative to the machine speed. The height at which the rice crop is cut at harvest plays a major role in determining the grain yield, ensuring optimum performance in minimizing grain losses and optimizing grain quality [24]. Reference [25] also utilizes the following four parameters: reel index, the cutting height of the crop, the horizontal distance of reel from the cutter bar, and vertical distance of reel from the cutter bar, as factors influencing combine harvester header losses as stated in Figure 2. Combine harvester header loss results from cutter bar strokes, the height of the reel, reel peripheral speed, travel speed, the width of harvest, the height of cutting, crop moisture, height and density of crop, feed rate of the crop [26].

### 2.2. Threshing Losses

The term "threshing loss" refers to unthreshed grain that is left behind by the combine head and transported to the machine's rear via a straw rack. Mechanical threshing loss refers to grain loss caused by an inefficient rubbing action between the cylinder and the concave. Grain losses due to threshing are significantly influenced by the performance of the combine threshing [23]. Impact and rubbing in a combine dislodge grain kernels from the plant head. The speed of the rotor or cylinder, the type and spacing of the concave, and the hardware parameters all affect threshing. Typically, mechanical grain loss occurs due to grain threshing operations [27]. Combine threshing loss is dependent on feed rate,

cylinder/rotor speeds, and concave clearance as depicted in Figure 3, which is affected by crop moisture [28].

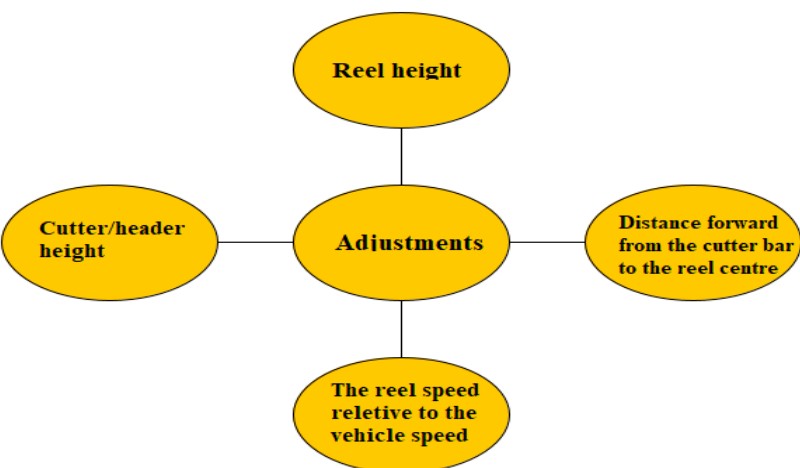

**Figure 2.** Machine parameters that affect/reduce combine header loss.

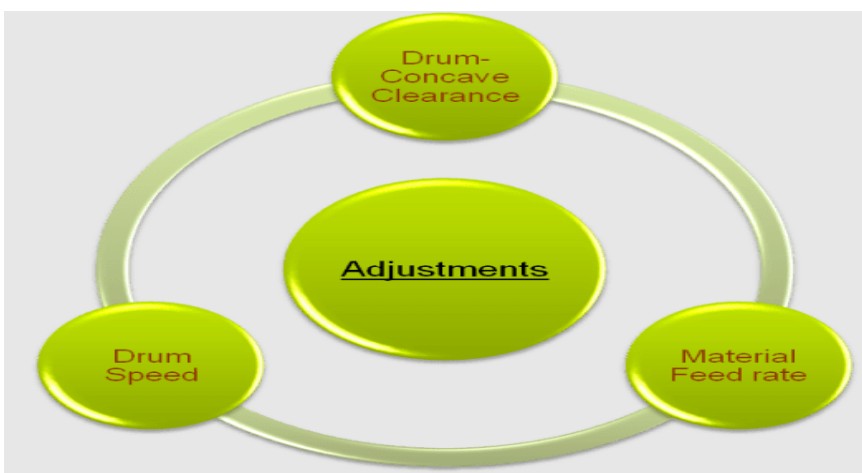

**Figure 3.** Machine parameters that affect threshing and separation losses.

### 2.3. Separation Losses

When evaluating a combine harvester's performance, grain separation losses are a critical parameter to consider [29]. Losses due to separation are the quantity of free grain that does not pass through the grates of the separation section in an axial flow [5]. Some changes that increase threshing efficiency will negatively affect separation efficiency. When the moisture content of the straw is low, the cylinder breaks the straw finely, allowing more material to pass through the sieve resulting in a separation problem. Additionally, Crop conditions, machine settings, and operator choices that affect separation efficiency have much in common with the threshing system.

### 2.4. Cleaning Loss

Cleaning losses refer to the number of grains mixed in the screenings of a combine harvester, which is generally measured by the rate of cleaning loss [30]. Grains are blown out by airflow during the cleaning process, and the loss during cleaning is unavoidable [7]. Cleaning systems in combines utilize a combination of air blast to lift off chaff and straw and a shaking action to draw grain downward through the sieves while moving larger particles to the rear. Again, the moisture content of the harvested grain is a crop condition

factor that will influence performance [31]. Machine settings that most affect cleaning system losses include fan speed and sieve opening as highlighted in Figure 4.

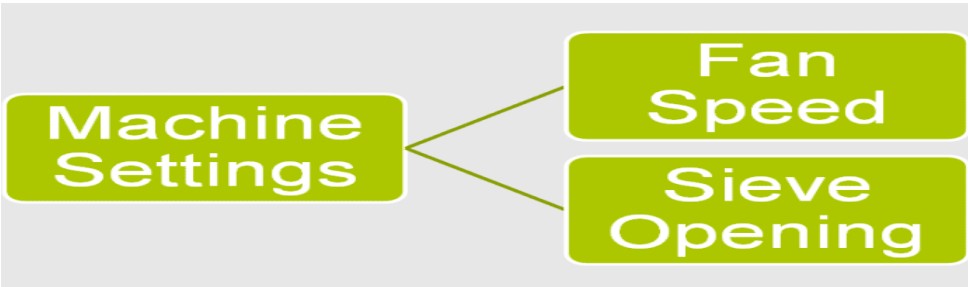

**Figure 4.** Machine parameters that affect grain cleaning losses.

*2.5. Parameters That Contributes to Grain Losses on Combine Harvester*

Determining appropriate harvesting parameters is an essential factor to consider a successful combine operation to minimize grain losses and enhance grain quantity and quality. Improper cutting height, combine forward speed, concave clearance, drum speed, fan speed, sieve opening, crop moisture content, reel speed, and horizontal and vertical distance of the reel from the cutter bar are all factors in combine grain losses. These factors affect the machine settings, field efficiency, and theoretical and effective field capacity.

According to [32], the following reasons contribute to the grain loss of this machine: improper cutting height, inconsistent placement of reel in respect to cutting bar, improper kinematic Index ratio, and inappropriate velocity of cutting bar or damaged blades. Reference [33] utilizes the combine forward speed; drum speed, crop moisture content at harvest, length of harvesting area, specific area for the harvested crop, and effective width as factors to estimate grain loss while operating a combine harvester. The results show that the drum speed substantially influences combine processing loss.

Reference [25] stated that the real index, the cutting height (mm), the horizontal distance of the reel from the cutter bar (mm), and the vertical distance of the reel from the cutter bar are all factors that affect header losses on combine harvesters. The combine harvesters reduce the amount and quality of rice grains, resulting in significant agricultural yield reduction. Concerning crop conditions and machine ground speeds, the biggest losses are generated by machines that have not been properly adjusted. Throughout the harvesting season, losses in rice output attributable to the usage of inappropriate machinery and procedures are estimated to be between 25% to 30% [10].

The results obtained during field experiments, according to [34], revealed that combine harvester performance is dependent on the cutter bar operating width, grain yield, and on the forward speed of the combine harvester, which is controlled by field conditions. The combine harvester's field speed is another element that affects grain losses and results in excessive grain loss during harvesting. According to [35], the most critical aspect in optimizing the performance of a combine harvester is field speed. It was reported by [36] that the incorrect travel speed of the combine harvester employed during the harvesting process was a contributing factor to paddy harvesting losses. Additionally, [37] state that the machine's forward speed is the primary element affecting the combine harvester's performance.

According to [38], farmer satisfaction with automated rice harvesting in Malaysian paddy fields was significantly influenced by the combine harvester's field speed. As a result, one of the major concerns during mechanical harvesting in Malaysian paddy fields is the field speed of the combine harvester.

From the research conducted by [38] at rice granaries in Bagan Serai, Perak state, Malaysia, the operator's daily field speed of the rice combine harvester was measured. Its impact on grain loss in paddy fields was also assessed. The study revealed that combine harvester field speeds ranged from 3.87 to 6.11 km/h in Malaysian paddy fields. The best

field speed was 3.87 km/h, resulting in 0.67% grain loss. The results are consistent with those obtained by [38], and the values are within the range of 3.0–6.5 km/h anticipated by the combined field speed (ASABE, 2011). Field speed is also correlated with grain loss. With a decrease in the forward speed, the amount of grain loss was significantly reduced. When a combine harvester's field speed is set too high, it increases field efficiency but decreases material capacity due to grain spill out on the ground. Because of this, optimum combine field speed can help prevent grain loss [38].

The major loss is focused on automatic harvesting with incorrect adjustment and calibration of the harvester's components. So the farmer ignores the relevance of these factors; also, a set of interaction factors that reduce harvester performance efficiency include the field type, crop variety, maturity, and angle of inclination [5].

### 3. Conventional Methods of Grain Loss Measurement

A variety of measurement methods exists to determine grain losses in combine harvesters. The existing conventional methods have a certain percentage of accuracy, but these methods are time-consuming, tedious, and labor-intensive. From the literature, header losses are collected from the portion of ground protected from combine efflux by using rolls of cloth. The loose grains and complete and incomplete ear heads that fall on the marked area of the rice field after the harvesting operation by the combine are collected manually. This loss is due to an inappropriate cutter bar operation. To measure the threshing and separation losses, the straw and chaff efflux is collected separately by the two rolls of cloth, 30 m in length and one-and-a-half times the width of the straw/chaff outlet [29]. As the sheet of the roll cloth unrolled, one sheet retained or collected the threshing loss and the other one from the sieve, 20 m run length, collected the separation loss. The unrolling operation starts in advance and terminates 5 m before the end of the swath. According to [18], the combine was stopped during harvesting and moved back up to 15–20 ft. A quadrant (1 ft2) was laid out in at least four places at random along the header. The grains collected from those places are measured as "header losses". A similar method can also be used for measuring threshing losses behind the combine.

Reference [9] measured the harvest losses (threshing and separation) randomly selected within the harvested area at twelve different places. The area was 0.5 m in the direction of combined travel, using a 50 cm × 50 cm wooden frame. All the threshed and unthreshed grain that fell into the frame after the machine ran over it was picked up manually. All the grains and panicles inside the frame were collected and weighed as grain losses.

Reference [10] measures header losses during harvesting by allowing the combine harvesters to move forward for 50 m before stopping. The header unit was raised, and the machine was backed 5 m. The grains and panicles were manually picked up from the 1 m$^2$ quadrate placed in front of the parked machine as shown in Figure 5. The header losses were determined by weighing the fallen grains and panicle grains collected.

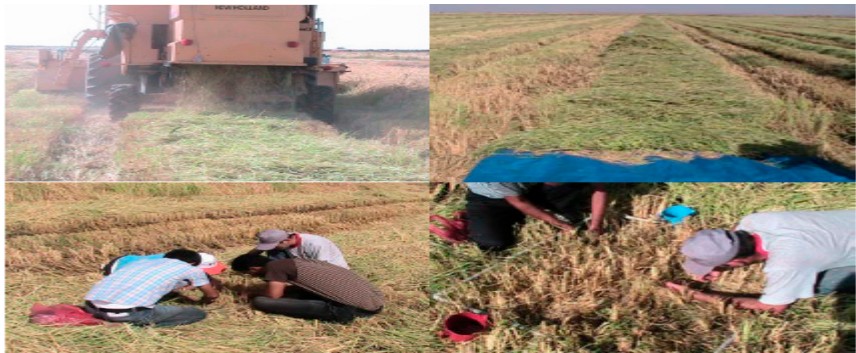

**Figure 5.** Manual grain loss sample collection [10].

Using the method of three-quarter square frames as indicated in Figure 6, the losses are taken with frames (50 cm × 50 cm) of a quarter square meter. When placing the frames, the crop divider trace is found first. The frames are placed at the location of the trace to determine the harvest losses. To find the losses, all the ears collected from inside the frames are measured and recorded as grain losses [10].

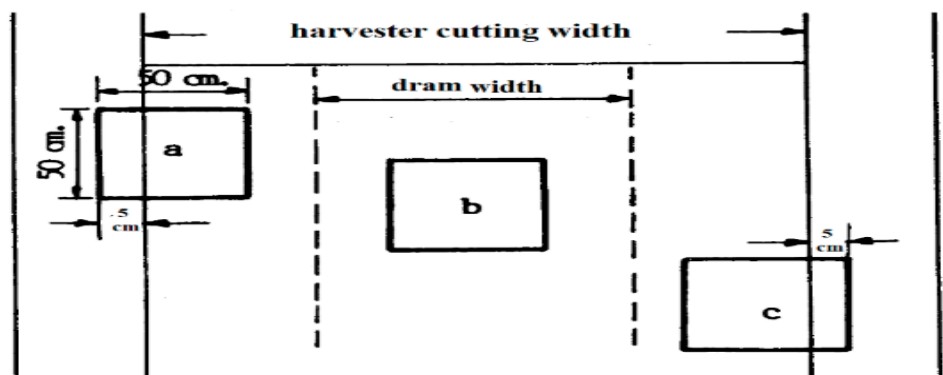

**Figure 6.** Frame sampling location for three quarter square frame [10].

A frame was placed over each row behind the combine as indicated in Figure 7, and the number of loose grains on the ground within the frame was collected. The number of grains that are still attached to the threshed cobs is measured as "cylinder loss". The frame was also placed in front of the combine where the separator had not passed, and the loose grains collected inside the frame were counted as header loss [16].

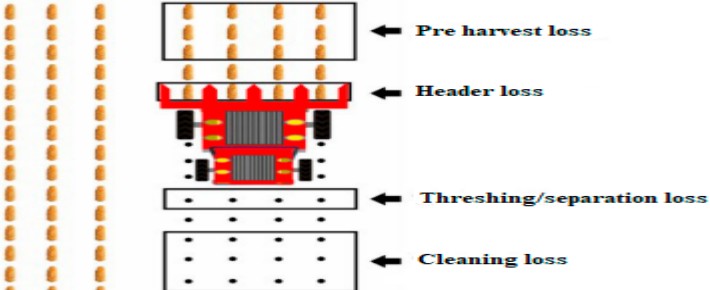

**Figure 7.** Measurement methods of grain harvest loss [16].

In a study by [28], the combine harvester losses were measured using the Embrapa method, where losses behind the combine were measured when the machine was running in the field at full capacity and at normal operating speed. The losses were measured across the full width of the combine header in a rectangular pattern of sufficient length to provide a 2.0 m$^2$ area as shown in Figure 8. All the losses on the ground were picked up and the grain attached to the stalk. The collected grains were weighed and measured as grain losses.

In a study by [38], grain loss was specified as the total field harvesting losses due to combine harvester movement on the fields. A sampling process was conducted in the said sub-plots to measure the grain loss. Five quadrats with an area of 1 m × 1 m were placed randomly inside each sub-plot. Prior to collecting the grain loss, the sampled areas were marked. The sampled areas were cleaned from natural grain loss before the machine entered the field. The average grain loss per plot was measured and calculated.

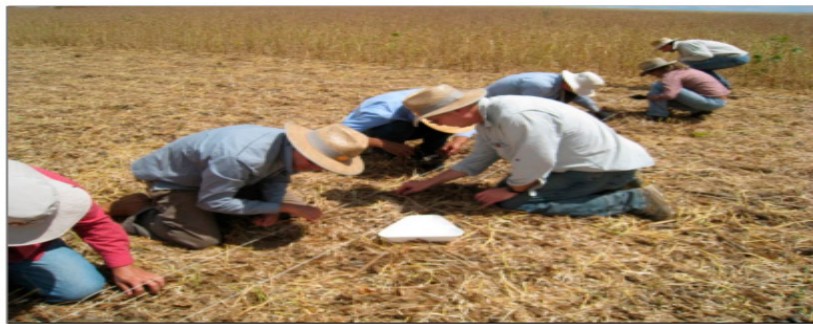

**Figure 8.** Picking grain losses in a 2-m$^2$ area [28].

A combine harvester header was lifted and reversed after reaching steady state speed under full load conditions. A quadrant frame (1 m × 1 m) was randomly placed along the cut swath in front of the machine, and grains were manually collected from within the frame and weighed as header loss [24]. The combine harvester was stopped during harvesting 100 feet from the row ends. A wooden frame or four wooden stakes and strings were placed along the 100-foot swath width. The combine was backed up 20 feet, and the kernels in the 10 square foot plots over each row in front of the combine were counted as displayed in Figure 9. The collected grains on the field were measured as combined field operation grain losses [39].

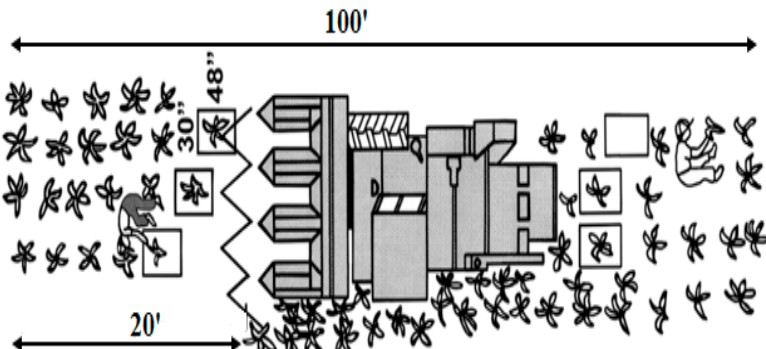

**Figure 9.** Measurement of grain loss by backing up 20 feet in a 10 square foot frame [39].

In addition, the ear and head losses were measured ahead of and behind the combine harvester in 1/100 of an acre for a given row spacing and swath width by placing a rectangular frame or marking with a stick and string as described in Figure 10. After the combine passes across the mark areas, the grains collected in those areas are the ear and header losses.

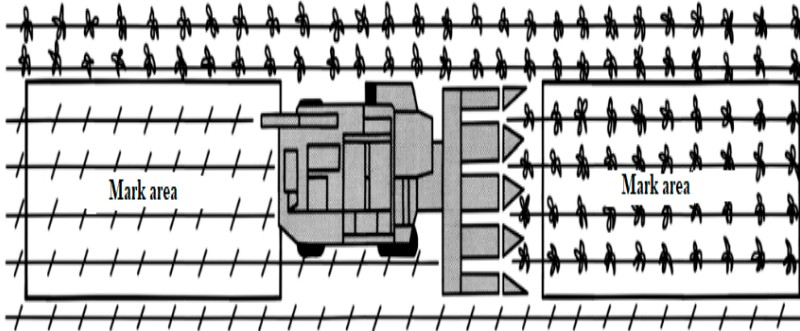

**Figure 10.** Measurement of ear and header losses from the end and front of the combine [39].

After the combine had passed, a 4 m$^2$ steel frame was placed at ten different locations in the field. The shattered grains and exited material from the combine were gathered from

the enclosed area of the frame. The samples so gathered were threshed, winnowed, cleaned, weighed, and recorded as harvesting and threshing losses by combine [40].

Reference [41] employed the continuous method of combine harvester grain loss assessment described by [42]. In this method, two woven plastic sacks were fitted at the rear of the machine in such a position that one sack caught the concave efflux and the other straw walker efflux as presented in Figure 11. The sacks were fixed at the start of 20 m distance; at the end of the swath distance, the sacks were removed, and the machine was stopped. The grains were separated from the chaff and straw manually. The grains from the concave efflux are the threshing loss, while those collected at the straw walker are measured as the separation loss.

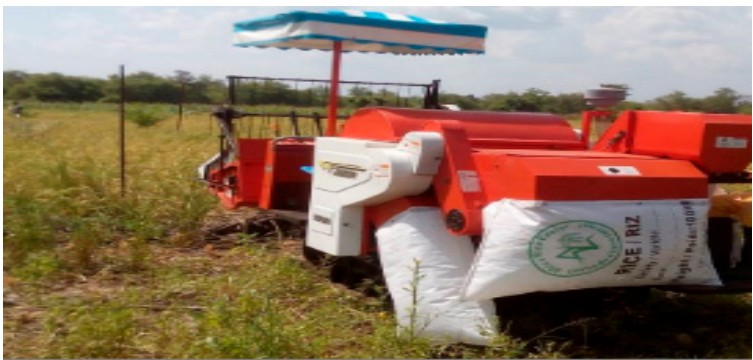

**Figure 11.** Collection of grain losses in a woven sack at the rear and concave of the machine [41].

In order to measure the losses at the header of the combine, the machine was driven in the field and, after achieving a steady-state condition, it was stopped. Then the combine was back up a distance equal to the longitudinal distance between the cutter bar and the discharge chute. The sample area was marked off in front of the combine, and the losses collected from the area were weighed as the header loss [43].

## 4. Grain Loss Sensing Method

Sensing technology is a method that is presently automated, remote, and prospective. It is used to monitor grain quality and identify losses. It provides an opportunity to deploy and install capabilities more quickly [44]. The word "sensor" refers to a device or system that responds to a physical or chemical quantity to generate an output that is a measure of that quantity (e.g., gas concentration, ionic strength measurement) [45]. A grain loss sensor is an important monitoring element in a combine harvester [26]. Sensors have a wide range of possible functions in grain loss monitoring. When utilized appropriately, grain loss sensors may be one of the most effective instruments for assisting in determining how effectively a combine is operating [46]. Grain loss monitors are instruments installed on combine harvesters that allow grain loss to be measured on various parts of the combine [33].

## 5. Related Studies on Grain Loss Sensor Monitoring

In a study by [47], collision signals were processed using a YT-5 piezoelectric ceramic sensor, which was chosen because of its high voltage amplitude and short rise time. In the end, a mathematical monitoring model was developed after a thorough comparison of the feeding quantities. Consequently, a grain impact sensor prototype was assembled on experimental test bench in a laboratory as presented in Figure 12, further installed on a combine harvester, and utilized in field experiments. A mathematical monitoring model was used to verify the validity of the method.

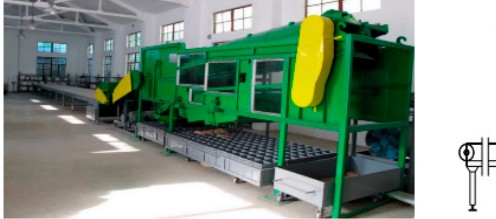 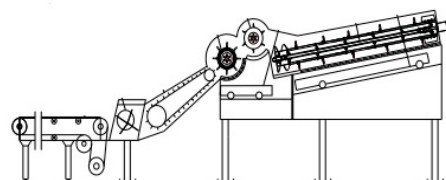

**Figure 12.** Experimental test bench for threshing-separation–cleaning [47].

In a study by [48], YT-5L piezoelectric ceramics, 304 stainless steel sensitive plate, and viscoelastic damping layer were used to develop the grain loss monitoring sensor. The grain loss-monitoring sensor was installed on the rear rack of the cleaning shoe under the sieve. A discrete element method (DEM) was used to study the collision behavior of the grains with the sensor. The signal processing circuit, which consists of the sensitive material, charge amplifier, band pass filter, and voltage comparative circuit, was designed to discriminate the full grains from materials other than grains (MOG). The performance of the sensor signal processing circuit was conducted on a calibration table in the laboratory and used on a combine harvester in the field.

According to [49], the novel sensor was built using six components: an LCL2218 PVDF piezoelectric film, a viscoelastic damping material, a stainless steel plate 304-impact plate, a fixed mount, a shielded wire, and a charge amplifier as illustrated in Figure 13. The PVDF piezoelectric film was attached to one side of the impact plate, while the fixed mount was attached to the opposite side. The PVDF piezoelectric film is mounted at the combine's exit via a fixed mount that regulates the installation angle and height of the novel sensor. Between the PVDF piezoelectric film and the impact plate is the dampening material. Finally, a kinematic model was used to optimize the sensor elements. The damping material and impact plate thicknesses were optimized using the principle of deformation energy.

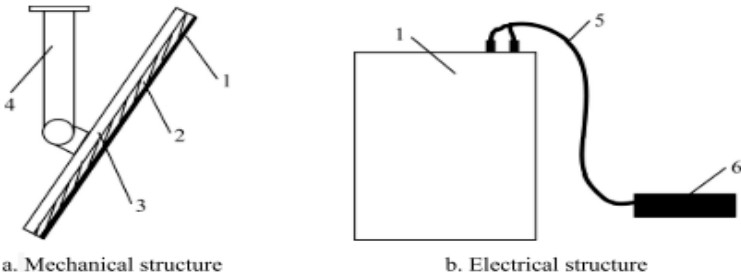

**Figure 13.** Structure of noble grain flow sensor. 1—PVDF piezoelectric film, 2—damping material, 3—impact plate, 4—fixed mount, 5—shielded wire, 6—charge amplifier [49].

In the center of the instrumented plate, a piezoelectric element YT-5L was installed. Rice grain was used in the laboratory for the impact tests. The instrumented impact plate was placed 300 mm above the test bench as shown in Figure 14. A charge amplifier processed the signals, and the voltage signal was monitored by a 500 kHz storage digital oscilloscope [50].

Reference [51] developed an array of the sensor using PVDF film, PET film, rubber foam damper, and aluminum plate. The signal conditioning circuit was made up of a PVDF sensor, charge and voltage amplifier, band pass filter, diode detection, an adaptive voltage comparison module, and counter. The system's software was designed using Keil C51 to develop the microcontroller unit (MCU) program. The experiment was carried out in a laboratory as demonstrated in Figure 15, and the fixed angle and sensor height were the same in both experiments. Finally, 12 tests were conducted for two grain varieties with two different moisture content.

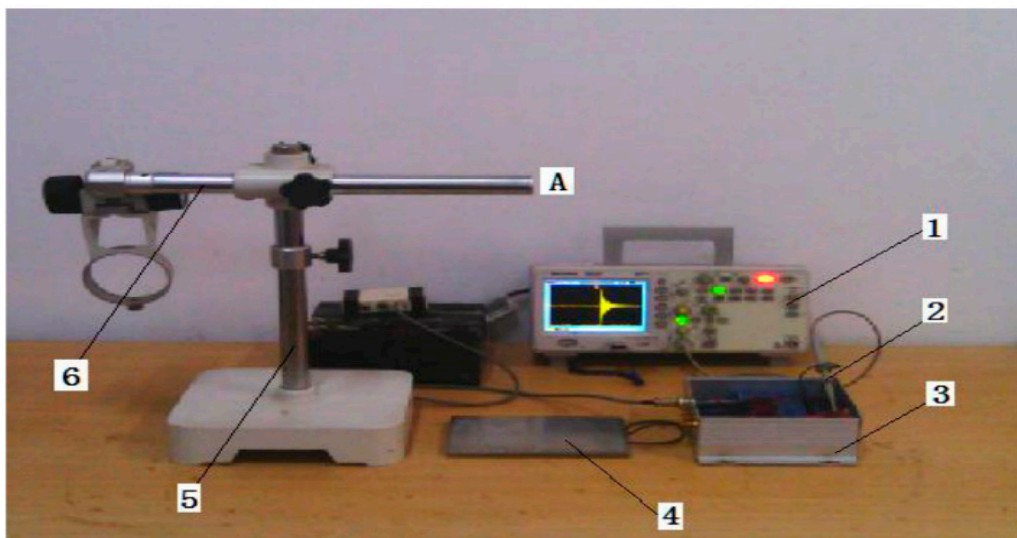

**Figure 14.** Single grain impact experimental system. A—grain release point, 1—digital storage oscilloscope, 2—probe, 3—signal processing circuit, 4—grain loss sensor, 5—support bar, 6—height adjustment bar [50].

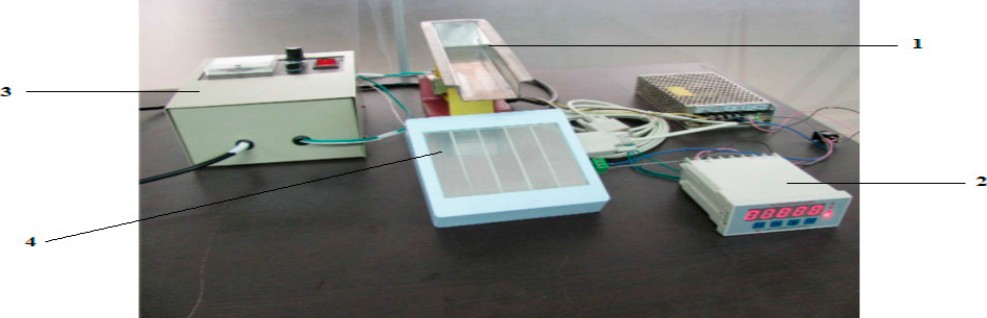

**Figure 15.** Installation of the experimental system. 1—grain conveyor, 2—display counter, 3—signal conditioning circuit box, 4—sensor platform [51].

Reference [52] developed a loss detection system using a PVDF piezoelectric film sensor, steel plate, and rubber foam damper to develop the sensor structure. The monitoring method adopted the discrete point method to establish the mathematical monitoring model. The monitoring system was evaluated or tested using a threshing and cleaning test bench as presented in Figure 16. Matlab was used to carry out polynomial fitting of the collected data. The experiment was conducted on a threshing and cleaning test table in a laboratory based on the 4LZ-10 wheat harvester developed by Shifeng Group. It is mainly composed of a threshing drum, a concave plate screen, a cleaning screen, and a fan. The optimum monitoring area of the sensor was found by studying the distribution law of grain quality of the clear separation loss, and the mathematical model of the monitoring of the cleaning loss was established.

Reference [53] developed a sensor structure using 2-crossed PVDF film, PET film, silicon rubber, and 304 stainless steel sensitive plate. The signal and control acquisition system was designed using the LabVIEW software program to send instructions to the micro controller and save the signals from the PVDF film. The performance of the sensor was done using a full factorial collision, continuous collision, and simultaneous collision test using the impact position and drop height as variables to verify the sensor performance on a test bench as shown in Figure 17a–d showing the major parts of the test bench.

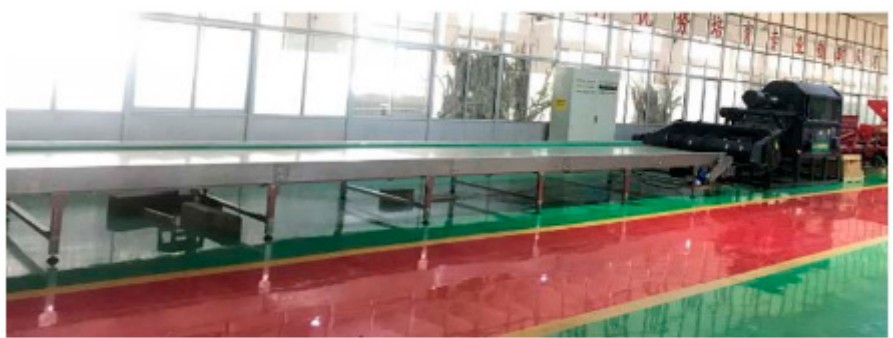

**Figure 16.** Experimental table for threshing and cleaning [52].

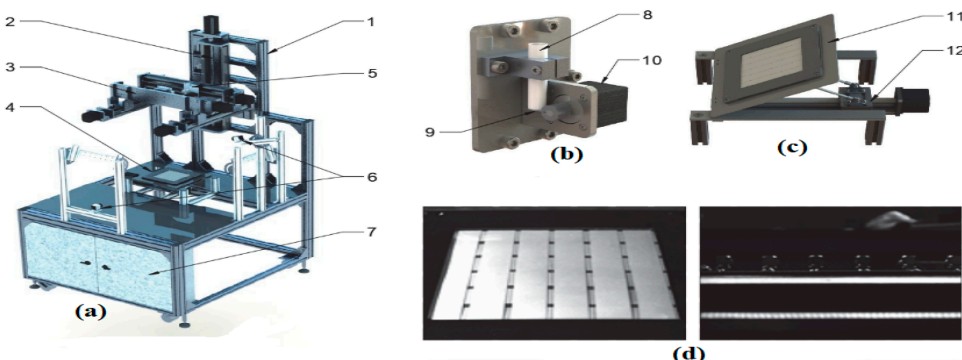

**Figure 17.** (**a**) Experiment test bench; (**b**) Grain release adjuster; (**c**) Sensing plate slide adjuster; (**d**) Sensing elements (1—support, 2—height adjustment mechanism, 3—position adjustment mechanism, 4—angle adjustment mechanism, 5—release mechanism, 6—high-speed camera, 7—control cabinet, 8—tube, 9—cover, 10—motor, 11—plate, 12—slide) [53].

With stainless steel 304 as the sensitive plate, the P5-3B piezoelectric ceramic was used. Rapeseed grain loss ratio is estimated by combining data from four monitoring devices situated in the tail sieve's lateral direction. An integrated sensor, signal processing circuit, and host computer constitute the monitoring systems. Filtering, full-wave rectification, and envelope comparison all fall within the ambit of the signal processing section. The host computer was developed based on the FPGA basic board. The developed mathematical model was used to calculate and display the real-time cleaning loss. For this experiment, the system was tested in a laboratory as shown in Figure 18. [54].

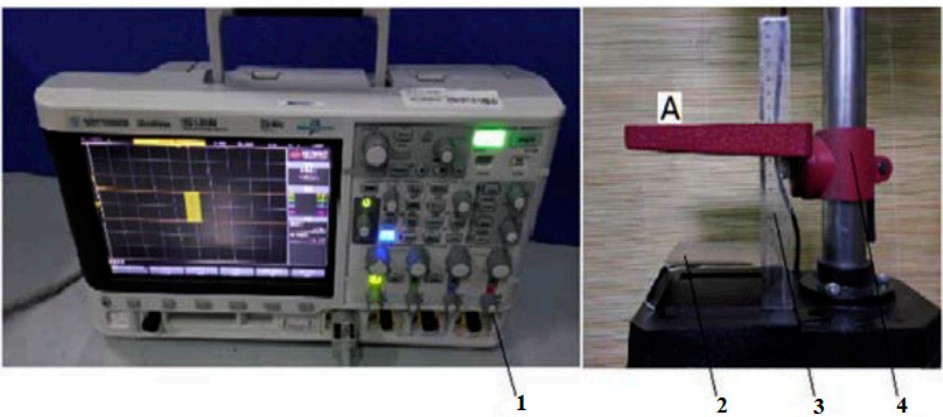

**Figure 18.** Grain impact experimental system: A—Grain release point, (1—oscilloscope, 2-sensor, 3—height gauge, 4—height adjuster) [54].

Reference [55] developed a grain monitoring system using a 2-sampling box with a visual window inside; a 3-CMOS camera installed inside the sampling box; a 4-Choke plate pushed and pulled by an electromagnet and a spring; a 5-Electromagnet and spring installed on the sampling box; a 6-control unit; a 7-interface; an 8-image processing unit; a 9-RGB monitor and a 10-PC as show cased in Figure 19. Grains are transported into the grain bin and then fall through the sampling box mounted beneath the grain bin auger. A camera mounted inside the sampling box was used to capture the kernels and impurity mixture images. The particles of kernels and impurities were classified by machine vision technology, and hence a result of the grain impurity ratio is computed and sent to the cab. In order to assess the accuracy of image processing, experiments were performed to collect samples, capture images, and process them in the laboratory.

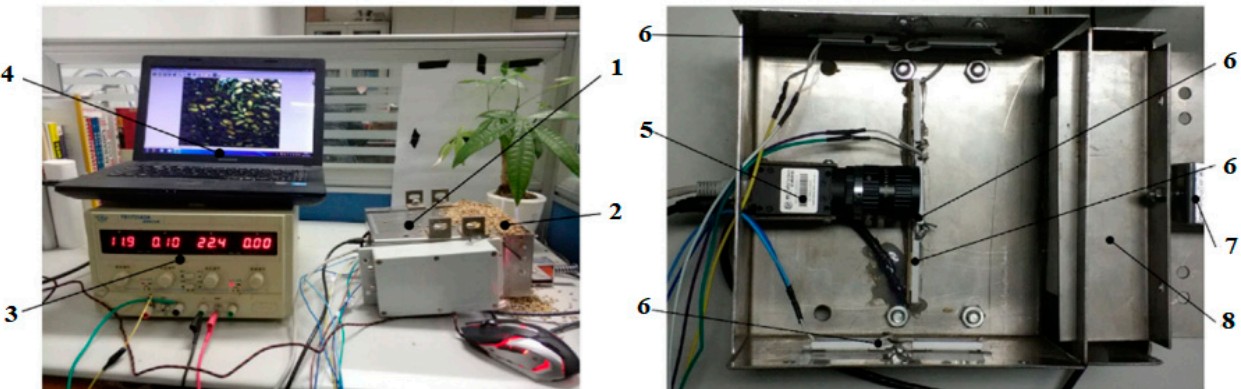

**Figure 19.** Installation of the experimental system. 1—sampling box, 2—rice grain, 3—stabilized voltage source, 4—PC, 5—CMOS camera, 6—LEDs, 7—electromagnet, 8—choke plate [55].

In a study conducted by [56], the sensor structure was designed based on a 5-layer structure; two PET layers, PVDF piezoelectric film, damping material, and an aluminum alloy base layer as a sensitive plate. The system includes a data acquisition module, a filtering algorithm, a wave-shaped module, and a wireless transmission module as presented in Figure 20. A Kilman filtering algorithm was used to process the signal output from the amplifier to eliminate external mechanical vibration during the cleaning process. In order to guarantee the data processing speed of the MCU, the 2 × 2 array was adopted in this phase to develop the basic system. The experiment was conducted on a test bench in a laboratory.

Photoelectric film pasted on a sensitive plate was used as a sensing element to record the impact of grain and MOG. The software for the monitoring system was designed using LabVIEW. The data sets were machine learned using the decision tree algorithm to classify the data into a discrete form. The standard tree is represented by the 48 algorithms. In addition, to simulate the effect of materials falling from the concave screen when harvesting, a sensor calibration test bench was used as shown in Figure 21. To verify the validity of the developed monitoring system, the monitoring accuracy of the system "with recognition model" and "without recognition model" were compared at the grain/impurity ratio levels [7].

Table 1 shows an overview of existing grain loss sensing monitoring systems from previous studies, including the types of sensors utilized, where they were installed, and how accurate they measured grain loss.

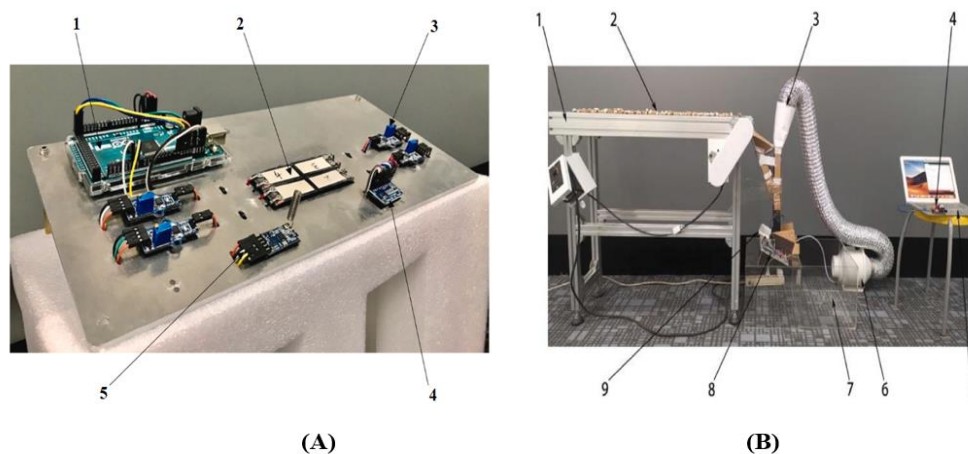

**Figure 20.** (**A**) The layout of the sensor impact; 1—MCU, 2—sensor, 3—amplifier, 4—A/D converter, 5—wireless transmission. (**B**) Experimental platform; 1—conveyor, 2—grain and residue, 3—air duct, 4—data receiver, 5—data processing, 6—fan, 7—recycling box, 8—impact plate, 9—power bank [56].

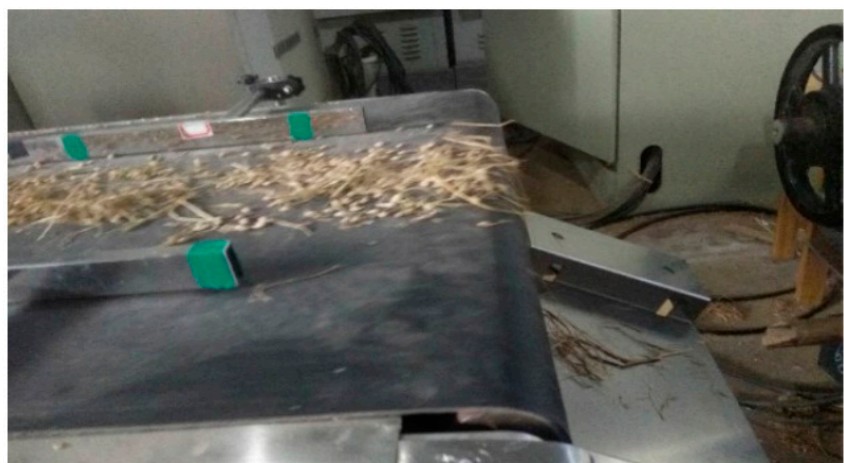

**Figure 21.** Experimental test bench of grain/MOG falling on the sensor plate [7].

**Table 1.** Review of the existing grain loss sensors and measurement errors.

| Authors | Type of Losses | Sensor Used | Sensor Position | Measurement Error (%) |
| --- | --- | --- | --- | --- |
| [7] | Cleaning loss | PVDF film sensor | Under the concave screen of the lab test | 0.7 highest error, 1.3 lowest |
| [21] | Separation/cleaning | Piezoelectric ceramic | Under the sieve | ≤3.46% |
| [29] | Separation loss | Piezoelectric ceramic | End of the walker | 4.63 |
| [48] | Cleaning loss | Piezoelectric ceramic | Under the cleaning sieve | 4.48 |
| [50] | Separation loss | Piezoelectric ceramic | Under the threshing sieve | 3.83 |
| [51] | Separation/Cleaning | PVDF film sensor | Under the cleaning/separation sieve | Rice 2.5, Wheat 4.8 |
| [52] | Cleaning loss | PVDF film sensor | Below the threshing and cleaning sieve | 1.8–3.15 |
| [53] | Grain losses | Two crossed PVDF film | Test bench in the laboratory | Not specified |
| [54] | Cleaning loss | Displacement sensor | Under the cleaning sieve | Not specified |
| [56] | Cleaning loss | PVDF film sensor | Test bench in the laboratory | Reduced from 12 to 3 |
| [57] | Cleaning loss | PVDF film sensor | Rear of the vibrating cleaning sieve | Not specified |
| [58] | Separation loss | PVDF film sensor | Under threshing/separation rotor | 3.40 |

## 6. Principle of Grain Loss Monitoring System

The principle of grain loss monitoring (measurement) using sensors as presented in Figure 22 occurs when a grain hits the sensitive plate and sends an impact that causes vibration. The sensor, which is attached to the sensitive plate, transforms the vibration

into an electric signal, and the corresponding signal processing circuit processes the grain impact (collision) signal into a waveform for counting, while the signals for the material other than grain (MOG) are filtered out. The signal representing the number of grain impacts is sent to the cab at a fixed interval. The in-cab monitor interprets the signal and provides a readout for the operator, usually as a graph or the accumulated grain number.

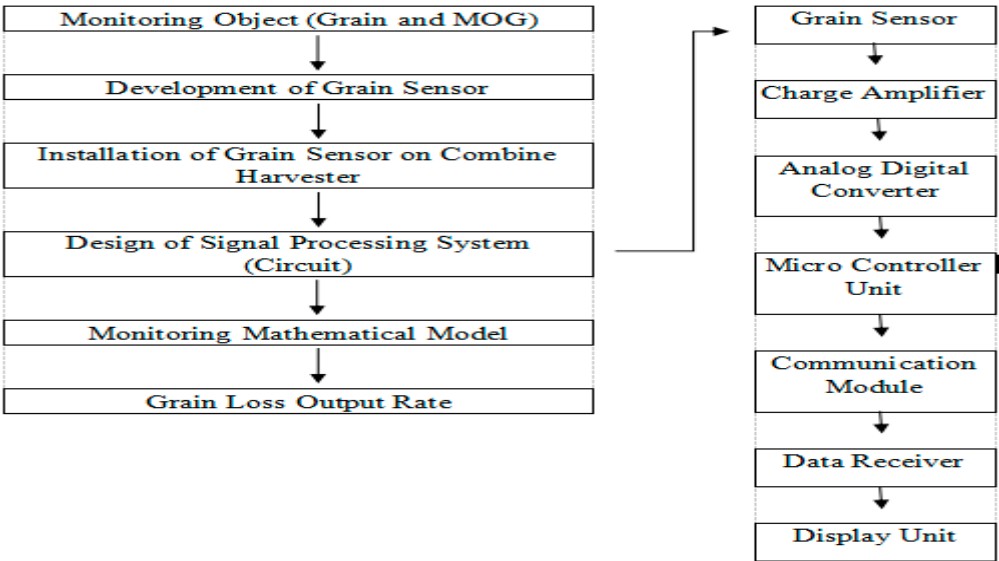

**Figure 22.** Schematic diagram of grain loss sensor monitoring process.

The summary of the various types of sensors, as well as their benefits and disadvantages, as found in previous studies, is presented in Table 2.

**Table 2.** Types of sensors used for grain loss detection.

| S/N | Type of Sensor | Critical Point (Comments) | Photo |
|-----|----------------|---------------------------|-------|
| 1 | Piezoelectric ceramic sensor | Because of the brittleness of the sensor material, it is susceptible to breakage due to vibration and has a lower level of reliability than the other materials. The vibration interference of the combine reduces the sensor's measurement accuracy [48]. | |
| 2 | Acoustic sensor | The sensor's sensitivity severely diminished due to the mechanical vibration of the machine [56]. It has a limited degree of precision [57]. | |
| 3 | PVDF piezoelectric | It is the most commonly utilized grain loss sensor. It has a negligible effect on the structure of the testing system. It also offers a considerable advantage in terms of sensing components [57]. | |

**Table 2.** *Cont.*

| S/N | Type of Sensor | Critical Point (Comments) | Photo |
|-----|----------------|---------------------------|-------|
| 4 | Force sensor | The sensor can only detect a fraction of the discharge grain and the surface of the sensor is very small to be installed on the combine harvester [59]. |  |
| 5 | Pressure sensor | The conventional pressure sensor is incapable of distinguishing between grain and residue. Additionally, the sensor's response time is very lengthy [60]. |  |
| 6 | Piezoelectric crystal sensor | This type of sensor increase not only the signals of the grain cleaning loss but also provide consistent sensitivity for the sensitive element stable performance, and high signal to noise ratio were the key characteristics of the sensor [61]. |  |

## 7. Limitations of the Existing Grain Loss Sensor Monitoring Systems

The following limitations were identified in the reviewed studies in order to improve and optimize the currently developed grain loss sensor monitoring systems:

1. The grain loss sensor has a low resolution. It should be upgraded to meet rice harvesting requirements, which indicate that total grain loss must be less than 3% [50].
2. Due to the dense and concurrent collisions of the materials, the impact signals overlap [53]. The sensor's integration and protection are required to withstand the rigorous operating conditions encountered in field applications.
3. The amplifier's output signal was still irregular due to interference signals caused by external mechanical vibration. To process the signals, it is necessary to improve the filtering module [20].
4. The separation of grains from materials other than grains (MOG) must be improved because grains are identified as residues in some of the developed grain sensor monitoring systems, and residues are also identified as grains [56].
5. Interference between signals generated during field harvesting is unavoidable due to machine vibration disturbance, the low recognition accuracy of grain impact signals from materials other than grains (MOG), and grain moisture content change [7].
6. Finally, it is illustrated that most currently developed grain loss monitoring systems have been tested in the laboratory, but only a few were implemented in field harvesting.

## 8. Conclusions

Sensing technology enables an automated technique that is quick, promising, and precise for reducing grain losses in combine harvesters. According to the review, grain loss sensor monitoring is more effective and efficient than conventional approaches, which are time-consuming and labor demanding. There are limited studies on grain loss monitoring at the combine header, and hence additional research in this area is recommended. One of the present grain loss sensor monitoring systems' limitations is that their outputs display bar graphs or numerical data without a unit of measurement, making it difficult for combine operators to quantify the losses adequately. As a result, converting the existing grain loss sensor readings to meaningful quantities of absolute grain loss will enable combine

operators and farm managers to make improved economic decisions and control grain losses more effectively during harvest.

**Author Contributions:** Conceptualization, N.M.N. and M.I.B.; methodology, validation, formal analysis, N.M.N., M.I.B., S.A.A. and M.S.M.K.; Resources, N.M.N., M.I.B., S.A.A. and M.S.M.K.; writing review—original draft preparation, M.I.B.; editing, N.M.N., S.A.A. and M.S.M.K.; supervision, N.M.N., S.A.A. and M.S.M.K.; project administration, M.I.B.; funding acquisition, N.M.N. All authors have read and agreed to the published version of the manuscript.

**Funding:** This research was funded by the Ministry of Higher Education of Malaysia under Transdisciplinary Research Grant Scheme (TRGS/1/2020/UPM/7).

**Institutional Review Board Statement:** Not applicable.

**Informed Consent Statement:** Not applicable.

**Data Availability Statement:** Not applicable.

**Conflicts of Interest:** The authors declare no conflict of interest.

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
