# Peer review of "Sensing Technologies for Measuring Grain Loss during Harvest in Paddy Field: A Review"

_agriengineering, doi:10.3390/agriengineering4010020_

Round 1

Reviewer 1 Report

Line 63: correct the citation format

Line 96: the authors should mention the method of gathering information for the review, what was the methodology?

Line 128:  correct the citation format

Line 210: correct the citation format

Line 227 and 228: Reference missing?

Line 475 and 479: correct the citation format

An interesting review, but more focus should be given to precision technologies. The review should include what methodology was used to develop the review

Author Response

the responses have been updated

Reviewer 2 Report

The work concerns the review of the technology of grain loss detection during
combine harvester rice harvest process.
The authors focus on automated and remote methods using real-time sensors.

It is relevant to the topic covered in the journal

The mauscript is easy to read and written in fairly correct English.

L128: Jarolmasjed et al., 2013, 
L385: Sun et al., (2019) should be corrected into the citation format of the AgriEngineering Journal.

Please improve the clarity of the figures. They are blurred, especially Figures 1,2,10.

Entries 37 and 38 in the reference list are the same. Please check references

I miss a description of how the methods of monitoring grain losses during harvest improve the efficiency of the harvest. 
I am asking the Authors for comments on this topic.

Author Response

The responses have been updated

Reviewer 3 Report

The manuscript “AgriEngineering-1597070” entitled “Sensing Technologies for Measuring Grain Loss During Harvest in Paddy Field: A Review” by Nawi et al. deals with an interesting subject regarding the parameters that contribute to grain losses in combine harvesters, and conventional method of measuring grain losses as well as the various types of sensors used to measure and monitor grain losses, and issues concerning the design, operation, and limitations of sensing technology used in combine harvesting operations.

For publication in “AgriEngineering”, the topic and content are appropriate. The subject of the review is interesting and topical, with high scientific and practical importance. The introduction is in accordance with the subject and correctly presented. Numerous scientific articles of recent date and in concordance to the topic of the study were consulted. The methodology of the study was clearly presented, and appropriate to the proposed objectives. The obtained results have been fully analyzed. The scientific literature, to which the reporting was made, is recent and representative in the field. The editing and linguistic quality are good. In addition, it is easy to follow by the reader, the tables give good summaries and the text editing in a thoughtful conclusion part. However, there are some points that need attention in order for the article to be published. I would like to recommend the publication of this article, and a minor revision is required for the reasons listed below:

  • Keywords: Please change some keywords. Title and keywords must not contain the same words.
  • Be consistent with the formatting of references and cross-references. This has to be standardized across the paper.
  • Table 2: The quality of photos is very low and some of the photos are blurred. Please fix this problem.
  • Figure 16, 17, 19, and 20: Please refer to the sources of the photos in the figure’s title.
  • Lines 501-559: Authors should correct the form of references, as in the journal’s “Instructions for authors”.
  • Finally, the back matter section (author contributions, funding, etc.) is missing. The reviewer recommends the authors will carefully revise their manuscript according to the “Instructions for authors”.

Thank you for your consideration.

Author Response

The responses have been updated

Reviewer 4 Report

This paper is a review of the existing literature on grain losses during mechanized harvesting, presenting the sources of grain losses, the parameters that influence them, and the existing monitoring systems and sensors for grain loss detection and prediction.

In my opinion, this paper is original and interesting and I enjoyed reading it.

I think that the next papers are close to the subject of this review:

1. https://doi.org/10.31274/icm-180809-36 
2. https://doi.org/10.1016/j.procs.2017.11.426

Author Response

The reviewer stated the paper is original and interested